# Use of Artificial Intelligence in the Classification of Elementary Oral Lesions from Clinical Images

**DOI:** 10.3390/ijerph20053894

**Published:** 2023-02-22

**Authors:** Rita Fabiane Teixeira Gomes, Jean Schmith, Rodrigo Marques de Figueiredo, Samuel Armbrust Freitas, Giovanna Nunes Machado, Juliana Romanini, Vinicius Coelho Carrard

**Affiliations:** 1Department of Oral Pathology, Faculdade de Odontologia, Federal University of Rio Grande do Sul (UFRGS), Porto Alegre 90035-003, Brazil; 2Polytechnic School, University of Vale do Rio dos Sinos—UNISINOS, São Leopoldo 93022-750, Brazil; 3Technology in Automation and Electronics Laboratory—TECAE Lab, University of Vale do Rio dos Sinos—UNISINOS, São Leopoldo 93022-750, Brazil; 4Department of Applied Computing, University of Vale do Rio dos Sinos—UNISINOS, São Leopoldo 93022-750, Brazil; 5Oral Medicine, Otorhynolaringology Service, Hospital de Clínicas de Porto Alegre (HCPA), Porto Alegre 90035-003, Brazil; 6TelessaudeRS, Federal University of Rio Grande do Sul (UFRGS), Porto Alegre 91501-970, Brazil

**Keywords:** oral cancer, oral mucosa lesions, automated classification, computer vision systems

## Abstract

Objectives: Artificial intelligence has generated a significant impact in the health field. The aim of this study was to perform the training and validation of a convolutional neural network (CNN)-based model to automatically classify six clinical representation categories of oral lesion images. Method: The CNN model was developed with the objective of automatically classifying the images into six categories of elementary lesions: (1) papule/nodule; (2) macule/spot; (3) vesicle/bullous; (4) erosion; (5) ulcer and (6) plaque. We selected four architectures and using our dataset we decided to test the following architectures: ResNet-50, VGG16, InceptionV3 and Xception. We used the confusion matrix as the main metric for the CNN evaluation and discussion. Results: A total of 5069 images of oral mucosa lesions were used. The oral elementary lesions classification reached the best result using an architecture based on InceptionV3. After hyperparameter optimization, we reached more than 71% correct predictions in all six lesion classes. The classification achieved an average accuracy of 95.09% in our dataset. Conclusions: We reported the development of an artificial intelligence model for the automated classification of elementary lesions from oral clinical images, achieving satisfactory performance. Future directions include the study of including trained layers to establish patterns of characteristics that determine benign, potentially malignant and malignant lesions.

## 1. Introduction

The health field has undergone significant changes in recent years with the incorporation of technological advances [1]. The use of artificial intelligence (AI) has generated a significant impact, improving diagnostic accuracy and workflow in various areas of health and demonstrating benefits from patient triage to treatment [1,2,3].

In this context, its application in the classification of disease-related, usually patterned, images for diagnostic purposes has achieved promising results in different areas of health such as dermatology [4], diagnostic evaluation of head and neck cancer [2], cardiology [5] and neurology [6]. Convolutional neural networks (CNNs) are approaches that stand out in this scenario, being applied to the classification of images of conventional histopathology, confocal laser endomicroscopy, hyperspectral imaging [7] and radiology [2,8]. The CNN is a subtype of an artificial neural network that can be well applied to image classification. The CNN has a built-in convolutional layer that reduces the high dimensionality of images without losing its information [9,10].

The use of AI technologies could result in a significant impact in the field of oral medicine. The current scenario classifies oral cancer as an important public health problem [7,11]. Often, its diagnosis is established in advanced stages, leading to difficulty in controlling the disease and, consequently, resulting in a high mortality rate. This reality has remained unchanged in recent decades despite numerous research efforts to improve the available therapeutic approaches [1,12]. In a considerable number of cases, this disease is preceded by premalignant lesions (potentially malignant disorders), which can be identified from visual inspection of the mouth [1,13,14,15]. Among these lesions are leukoplakia [16,17,18], erythroplakia [19,20], erythroleukoplakia, proliferative verrucous leukoplakia, oral submucous fibrosis, and oral lichen planus [15].

The potentially malignant disorders are important lesions, which can often be underdiagnosed, as they are not associated with symptoms and present a pattern of heterogeneous clinical presentation [20,21]. Trained health professionals could, in theory, favor early diagnosis via screening [21,22]. However, there are few specialists [15,21] and many dentists feel insecure to carry out the evaluation and decision-making on the conduct [3].

AI algorithms incorporated into telemedicine and supported by smartphone-based technologies for the early screening of oral lesions can serve as effective and valuable auxiliary methods to decrease the cancer diagnostic delay [13,23]. Although the objective of clinical examination of the oral cavity is the identification of asymptomatic individuals who have certain diseases, other identified benign abnormalities may require biopsy and treatment [24].

Recent studies have investigated the feasibility of using AI to classify oral lesion images. Using residual networks, Guo et al. achieved an accuracy of 98.79% for distinguishing normal mucosa from oral ulcerative lesions [25]. Satisfactory performances were also shown via means of deep learning technologies on confocal laser endomicroscopy images [7] and deep neural networks [16,26] to discriminate between benign lesions, oral potentially malignant disorders, and squamous cell carcinoma. In the study by Fu et al., images of oral cancer with histological confirmation were used, located in different regions of the mouth and in various stages of evolution, with sensitivity of 94.9% [21].

The present study aimed to develop an AI model to classify oral lesion images from their main clinical characteristic (elementary lesion). AI models appear to be promising auxiliary tools for diagnosis and decision-making in relation to oral lesions. Their application in oral medicine is still critical due to the myriad of clinical manifestation of diseases in the oral cavity [15,21,26,27,28]. Moreover, there is a large paucity of databases to support the development of reliable algorithms [4,15,16,26,27,28]. Many studies have obtained remarkable results, but, in general, have focused on few pathological entities [29,30,31,32,33]. To expand the scope of the applicability of AI technologies in the clinical practice of oral medicine, the classification of the elementary lesion may be an important step in the process of building safe and effective diagnostic reasoning [34,35,36].

## 2. Materials and Methods

The method in this work follows the steps presented in Figure 1. First, the dataset was constructed and then the lesion region of interest was cropped. Then, these images were used to train the CNN. Finally, the CNN was able to predict the oral elementary lesions via an input image.

The details of each step are presented in the following sections. First is the dataset details, then the chosen CNN parameterization and finally the evaluation metrics.

### 2.1. Dataset

This research protocol follows the principles of the Helsinki Declaration, and was evaluated and approved by the local Ethics in Research Committee (GPPG nº 2019-0746) [37,38]. The study waived the use of a free consent form because the data are retrospective and anonymized and consist of image cavities, where it is not possible to identify individuals. No clinical or ethnic-demographic data were used [39]. 

The image selection period was from November 2021 to January 2022, conducted by a specialist researcher with 6 years of experience (first author of this work). A senior specialist (last author of this work) supervised and assisted in case of doubts.

A total of 5069 images of oral mucosal lesions from patients treated at the Stomatology Unit of Hospital de Clínicas de Porto Alegre and the Center for Dental Specialties of the Dentistry School from Federal University of Rio Grande do Sul were retrospectively recovered. These reference services receive patients referred from many municipalities in Rio Grande do Sul, the southernmost Brazilian state.

In these services, images are usually obtained as part of the outpatient care routine, predominantly using SLR cameras coupled with 100 mm macro lenses (aperture = 22, shutter speed = 1∕120) and circular flash at non-standardized conditions in relation to angulation, lighting and focal distance, and even other types of cameras such as smartphones. Images of lesions with different characteristics, with various levels of severity and dimensions ranging from millimeters to several centimeters, were included. All images used in this study were stored in jpg format.

The CNN model was developed with the objective of automatically classifying the images into six categories of elementary lesions: (1) papule/nodule; (2) macule/spot; (3) vesicle/bullous; (4) erosion; (5) ulcer and (6) plaque. Elementary lesions are a clinical classification used for soft tissue lesions. This classification is visual and clinical, without the necessity of complementary tests, thus serving as a gold standard for validation. Figure 2 presents a schematic representation of the elementary lesions and their respective actual examples to illustrate the classification categories.

Low quality images with low resolution, variations in angulation, use of cameras with various configurations, lighting variations, focus change, sub exposures or super exposures, which were all allowed. Images of different ethnic groups were also included to allow an approximation to the real practice scenario with their possible confounding factors. Representative examples of the image types considered are shown in Figure 3. Note the presence of good quality, low resolution, focus problem and illumination problem in the dataset images. The lack of a pattern in image sizes is also noteworthy.

In the labeling stage, the specialist researcher with 6 years of experience performed the manual delimitation and cropping of a rectangle encompassing the region of interest in all images. In the case of images with multiple lesions, each lesion was delimited and cropped separately, so that each input image to the CNN contained only one lesion (Figure 4A). Blurry images or images with overlapping lesions were excluded. In terms of the limits between the lesion and the clinically normal peripheral tissue, some more were closed and some broader crops were performed to include the lesion limits in the training set of the algorithm (Figure 4B). Only the cropped area of interest from the images was used in the training process, which was presented to the CNN in a labeled and supervised way. The senior specialist supervised and assisted in case of doubts. Reference anatomical structures were not considered, which in the oral cavity are complex and influenced by ethnic factors and dental treatments.

A total of 5069 images were used for the development of the model. For the training dataset, 70% of the images were used (3550 images) and the remaining 30% were used for the test dataset (1519 images). The dataset split was performed considering that different images from the same patient were used only in the training or in the test dataset, never in both.

### 2.2. Convolutional Neural Network

The model for distinguishing the elementary oral lesion was built using a convolutional neural network (CNN) (also known as ConvNet) framework, trained and tested using natural images. CNN’s convolutional kernel acts as a learning framework, reducing memory usage and computing faster [9]. CNNs have become popular by combining three architectural ideas: local receptive fields, shared weights and temporal sampling. These characteristics make networks excellent for pattern recognition [10]. The CNN mimics the way we perceive our environment with our eyes. When we see an image, we automatically divide it into many small sub-images and analyze them one by one. By assembling these sub-images, the image is processed and interpreted [10]. 

The CNN learns from the “back-propagation strategy” [9] and because of this the CNN works well with data that are spatially related [9]. For this reason, CNNs are widely implemented in application areas including image classification and segmentation, object detection, video processing, natural language processing and speech recognition [9]. The number of CNN layers necessary for each problem depends on the image complexity to capture low-level details. As the CNN deepens, the computational cost increases [9]. These architectures are composed of connecting 3-stage layers. The last stage of a typical CNN layer is responsible for the dimensional reduction and can be composed by either max. pooling or average pooling. This technique reduces “unnecessary” information, and summarizes the knowledge about a region [9].

While max. pooling returns the maximum value from the portion of the image paired by the kernel and works as a noise suppressant, the average pooling returns the average of the values from the same covered image portion defined by the kernel [9]. After the 3-stage layers, it ends up with a flatten layer. Usually for classification problems, we want each final neuron to represent the final class. The neuron behavior is connected to all previous layer activations, as a multilayer perceptron (MLP) [9,40]. Considering these connected layers, over a series of epochs the model becomes able to detect dominating low-level image features and classify the elements using the softmax classification technique [9].

Multiple connections learning non-linear relations can lead to overfitting, which is important to avoid as this generates false-positive classifications [9]. To avoid overfitting, the literature suggests data augmentation [40]. In addition to data augmentation, we proposed the reuse of existing learning. The ability to learn from a large number of experiences and export knowledge into new environments is exactly what transfer learning is about [40]. The transfer learning process also avoids overadaptation and underadaptation. It consists of using trained weights for a given problem *A* in the initialization of another network to solve a problem *B*. 

Learning transfer is especially important in the healthcare field due to the difficulty of accessing labeled and public datasets [41]. In the healthcare field, several techniques exist to improve performance, given these dataset limitations, which typically require more work and fine-tuning [42]. Typically, it is suggested to pretrain the weights of the classification network, by iteratively training each layer to reconstruct the images using convolutional or deconvolutional layers [42]. To address these limitations, there are two main kinds of learning transfer processes: (1)Initialize a model with weights: Usually recommended when the number of images is lower than 1000 per class. During the training, we keep some layers with fixed optimizing parameters in higher-level layers, once the initial layers have similar features regardless of the problem domain.(2)Retrain an existing model: Usually recommended when the number of images is larger than 1000 per class. An approach that most likely leads to overfitting may need more parameter optimization than the number of images available. Given the similarity with the dataset originally used for training, a large pretrained network can be used as a featurizer.


According to the description, the existing models were re-trained. The architectures were originally initialized with the weights trained using ImageNet [40]. By definition, we trained the whole network using our dataset. Table 1 presents the four CNN architectures selected in this experiment. Given the nature of the problem addressed in this study, we decided to test using ResNet-50, VGG16, InceptionV3 and Xception, which have been widely used in the literature to classify exams and are presented in Table 1.

### 2.3. Experiments

All of the experiments were executed using Google Colaboratory and GPU with 7.3 GB RAM. Four different notebooks were implemented to reproduce each of the different CNN architectures. Due to data limitation and the problem of having multiple classes, our experiments were composed of dataset preparation using data augmentation, CNN architecture implementation and the optimization of hyperparameters. 

#### 2.3.1. Data Augmentation

The data augmentation process consists of increasing the dataset via synthetic data based on the original dataset. The most common operations are rotation, translation, horizontal and vertical flip, subsampling and distortions. These operations intend to generalize the classification avoiding memorizing image regions containing the class [46]. However, some attention is needed to avoid generating noise in the training dataset. This process was especially important in this study due to the unbalanced dataset. We increased the number of images for the tree of the six classes, as presented in Table 2. 

In Table 2, the process to augment the dataset is summarized. Dataset increase was performed with the vesicle/bullous, erosion and macule/spot classes due to the low number of images. A specific Colaboratory document was created to generate the images based on rotation, zoom, width and height shifts and horizontal flips to reach approximately 1500 images for each category. For this augmentation, Image Dataset Generation via Keras was used and the new images were added to the original dataset. Then, we were able to connect the dataset storage with Google Colaboratory to use it in our experiments.

#### 2.3.2. CNN—Architecture Implementation

Four different notebooks were implemented in the selected architectures. This implementation was composed by the initialization using the default Keras implementation for these networks. The option was to initialize with ImageNet weights followed by the extraction of the top layer (dense layer composed of 1000 neurons). This process was replicated for each of the notebooks for a valid comparison between the networks.

Given the final layer extraction, we initially defined a dense layer with 5 neurons excluding the macule/spot due to the small number of images. After some experiments, promising architecture parameters were reached, such as the epochs for each network, optimizer, batch, image size and frozen layers. 

With the initial insights for the architecture decision, and powered by the data augmentation, the 5-neuron-dense layer was replaced by a new dense layer containing 6 neurons, each representing one of the following classes: vesicle/bullous, erosion, plaque, macule/spot, papule/nodule and ulcer. 

#### 2.3.3. Hyperparameter Optimization

The InceptionV3 architecture with a learning rate of 10^−6^, the categorical cross entropy loss function and the Adam optimizer were trained. Due to the low learning rate, the training was a fine-tuning of the original weights of the network. All input images were resized to 299 × 299 pixels. We achieved the best result with a final dense layer of 6 neurons and 50 epochs. These were the best hyperparameters identified compared with learning rates varying from 10^−3^ down to 10^−7^ and from 5 to 200 epochs. Testing a different number of frozen layers, we noticed that the best results were achieved by freezing all of the layers. Therefore, after several tests, the aforementioned configurations achieved the best results.

### 2.4. Evaluation Metrics

The confusion matrix was used as the main metric for the CNN evaluation and discussion. From the confusion matrix, it was possible to compute the true positives (TPs), true negatives (TNs), false positives (FPs) and false negatives (FNs) of the classification model. The accuracy (Equation (1)), precision (Equation (2)), sensitivity (Equation (3)), specificity (Equation (4)) and f1-score (Equation (5)) were used as evaluation metrics. These equations represent the classical evaluation metrics for the classification AI models.
(1)$$\text{accuracy}=\frac{\text{TP}+\text{TN}}{\text{TP}+\text{TN}+\text{FP}+\text{FN}}$$
(2)$$\mathrm{precision}=\frac{\text{TP}}{\text{TP}+\text{FP}}$$
(3)$$\text{sensitivity}=\frac{\text{TP}}{\text{TP}+\text{FN}}$$
(4)$$\text{specificity}=\frac{\text{TN}}{\text{TN}+\text{FP}}$$
(5)$$f1\text{score}=2\frac{\text{precision}.\text{sensitivity}}{\text{precision}+\text{sensitivity}}$$

## 3. Results

The classification using the standard metrics for the multiclass problem was evaluated. The oral elementary lesion classification reached the best result using an architecture based on InceptionV3. After hyperparameter optimization, more than 70% of correct predictions in all six lesion classes were reached considering our largest labeled dataset. Figure 5 presents the normalized confusion matrix with the model performance. Figure 5 indicates that our model reached a normalized true positive rate of 1 for vesicle/bullous, 0.79 for erosion, 0.87 for plaque, 0.94 for macule/spot, 0.79 for papule/nodule and 0.71 for ulcer. The average accuracy was 95.09%.

The data augmentation was successfully implemented and improved the classification model, once it resulted in a more balanced dataset and outperformed the macule/spot and erosion classifications, which presented poor performance in the initial experiments. The sensitivity and the specificity of each class classification were also evaluated and are presented in Table 3.

In the classification of the six oral elementary lesions, we achieved high sensitivity and specificity for four of the classes: vesicle/bullous, erosion, plaque and macule/spot. On the other hand, we achieved high sensitivity but medium specificity for two of the classes: ulcer and papule/nodule. Finally, in the overall analysis, an average sensitivity of 85.25%, specificity of 97.05%, precision of 86.32% and F1-score of 85.44% were achieved.

## 4. Discussion

This study aimed to develop an AI model based on a CNN, trained to perform the automated classification of images of oral lesions from their main clinical presentation (elementary lesions). This proposal was adopted since this is the first criterion applied in a flow of diagnostic reasoning by a specialist in his daily practice. At the moment when it is defined that an injury is characterized as an ulcer, a considerable number of diseases that present characteristically with plaques can be excluded [36,47], for example. Based on this assumption, the training of the architecture focused on the recognition of the elementary lesion as a way to organize a set of lesions into six distinct groups. Achieving success in this preliminary goal could be the first stage of developing an AI resource for differentiation between high- and low-risk oral lesions for malignancy, where other functionalities will be associated with contemplating this purpose.

The proposed model presented satisfactory performance when classifying the lesions, presented with an average accuracy of 95.09%, with high sensitivity (over 71%) and specificity (over 93%) for each category (Table 3). Analyzing the confusion matrix, we identified the high accuracy rates in the prediction of erosion, macule/spot and vesicle/bubble classes, which consist of classes of lesions that had a smaller number of samples and passed through the data augmenting process. Although this practice is widely accepted, it should be considered that this factor may have influenced AI performance. 

When analyzing the classification process according to clinical presentation, it is necessary to recognize that the manifestations of certain diseases vary throughout their clinical course. Squamous cell carcinoma, the most frequent malignant oral tumor, may be present as an ulcerated, erosive, plaque- or nodular-like lesion. In addition, it may be preceded by potentially malignant disorders [19,20]. Similarly, diseases that present as vesicles or as bullous in the oral cavity tend to rupture in a short period of time, starting to present as ulcerated lesions. Cases such as these show that AI resources that propose classifying diseases from clinical images should contemplate the possibility of inserting relevant clinical data that cannot be obtained from images, such as the evolution, duration or presence of symptoms [36,47].

We observed in our results a similar behavior between the classes of ulcer and nodule/papule, with misconceptions in the outputs of the model. Both classes presented lower sensitivity, with 71.71% for ulcer and 79.79% for papules/nodules and a high specificity rate, with 95.19% for ulcers and 94.58% for papules/nodules (Table 3). Seeking to understand these findings, the respective images were reviewed in the dataset. This exercise showed that certain ulcers resembled nodules with ulcerated surfaces. In addition, it was observed that ulcers with elevated edges can simulate a nodule. Despite the great potential to generate confusion, these images were included in the sample for clinical relevance, as we understood that creating experimentally ideal situations would limit the external validity of the results obtained.

Taking into account the complexity and anatomic variability of the oral cavity, classifying lesions from usual clinical images is a considerably challenging task [16,21]. To minimize artifacts and excess information common to this site, we delimited the region of interest containing the lesion to be analyzed. Thus, we eliminated confounding factors such as tooth overlap and anatomical variations such as oral mucosa, tongue, palate, lips and teeth. Nevertheless, we enlarged the clippings contemplating a margin on the periphery of the lesion, allowing the network to make connections taking into account the limits of the lesion and the transition tissue. Tanriver et al. agree that this approach favors increasing the number of images available for the experiment, as well as reducing information and confounding factors [27].

This study also used the semantic segmentation technique that, despite determining the limits of the lesion accurately, does not identify the variations in the lesion [16]. Welikala et al. argued that although the overall information provided by the image is important, the delimitation of the image of the lesion facilitates the decision-making of the model [26].

Within this proposal, the next step would be to improve the network by establishing patterns that point to being malignant or benign for each class of elementary lesions, without the need to attribute the definitive diagnosis of the pathological entity. This seems to be the more coherent path as opposed to seeking the definition of a final diagnosis, since there is a wide variety of diseases that present clinical aspects that overlap and depend on biopsy and histopathological examination to determine the diagnosis. From this approach that seeks to define being high or low risk for malignancy, recent studies have found promising results [16,21].

Although AI methods are becoming powerful auxiliaries in health diagnosis [2,13], many obstacles still need to be overcome [13,48]. Jurczyszyn et al. [30] developed their study with a sample of 35 patients, obtained following a protocol, and the author proposes the use of up to five texture characteristics of the oral mucosa to develop simple applications for a smartphone to improve the detection of leukoplakia of the oral mucosa [30]. However, the model seems to not be comprehensive, limiting the detection of oral leukoplakia.

The performance of CNN models is highly influenced by the quality of the images and the sample number [25]. The oral cavity represents an anatomical region difficult to access for image capture because it does not present natural lighting, it requires the removal of mucous membranes to visualize certain regions and there is the difficulty of standardization in angulation, distancing, framing and sharpness. Thus, we did not establish image capture patterns and admit variations in images with images with low resolution, sharpness, brightness and low quality for model development; yet, we achieved good performance.

The result of this work is promising since, in practical applications, the establishment of image quality criteria is a difficult process to generalize. Some models that have developed the architectures with strict quality controls highlight the importance of maintaining quality control in clinical application [6]. We demonstrated in this study that the performance was adequate despite using images of varying quality, without establishing any protocol of collection or configuration of a camera.

In the context of oral health assessment, screening is performed during dental care, considering the health history associated with visual and tactile examination [14,24]. In the future, our model should be improved with the inclusion of new layers, combining imaging modalities and the provision of clinical data to assist in the detection, classification and prognosis. This model should be in the form of a semiautomatic tool whereby the professional provides the image to the model and whereby it can give solid clues to the classification of a lesion.

In the same direction, Welikala et al. described an annotation tool designed to build a library of images of labeled oral lesions that can be used both for a better understanding of the appearance of the disease and the development of AI algorithms specifically aimed at the early detection of oral cancer. They are currently in the process of collecting clinical data with the inclusion of metadata such as age, sex and the presence of risk factors for cancer [26].

Many models prove the promising potential of computational techniques in the recognition and automated classification of oral lesion images. However, they have used standardized and high quality images such as hyperspectral images [45], autofluorescence [27], intraoral probes [49], histological images [31], fractal dimension analysis and photodynamic diagnosis [32] that are not widely used in clinical practice. In addition, some studies have demonstrated good results from a small and incomprehensive dataset, such as the detection of cold sores versus aphthous ulcerations using 20 images [4], cancer versus normal tissue with a sample of 16 images [33], leukoplakia, lichen planus and normal mucosa from 35 patients [30], and the diagnosis of tongue lesions from 200 images [50]. All of these studies demonstrate that AI has great potential in automated oral image classification, but many challenges remain, limiting generalization for working in real-life practice.

Despite obtaining good performance in the proposed model, we observed some limitations. When analyzing the images of the mouth, we identified situations in which more than one disease was present in the same area of interest. In this case, the network would need to consider subcategories contemplating the elementary lesions or provide two examples of output information for the same image. Another identified situation was related to lesions with heterogeneous clinical presentation, with lichen planus, for example, where in some cases we can identify the reticular, erosive and ulcerated pattern simultaneously in the same area of interest. To work with this limitation, modeling an algorithm that allows for the output of more than one classification could be an alternative.

This was the first step in the process of modeling architecture that has the potential for applicability in real-life practice and is able to make a positive impact as an auxiliary tool to support the process of the diagnosis of oral lesions for professionals without specific training in the area of stomatology, especially in the early diagnosis of oral cancer. 

## 5. Conclusions

Our proposal was to approximate the proposed model to the initial reasoning of the specialist when performing the differential diagnosis process in stomatology, developing a multiclass model based on the elementary lesion. Our results were promising and are part of a first step towards the development of a model capable of recognizing patterns in images. Further studies are needed to include layers trained to establish patterns of characteristics that determine benign, potentially malignant and malignant lesions, individualized for each category of elementary lesion.

## Figures and Tables

**Figure 1 ijerph-20-03894-f001:**
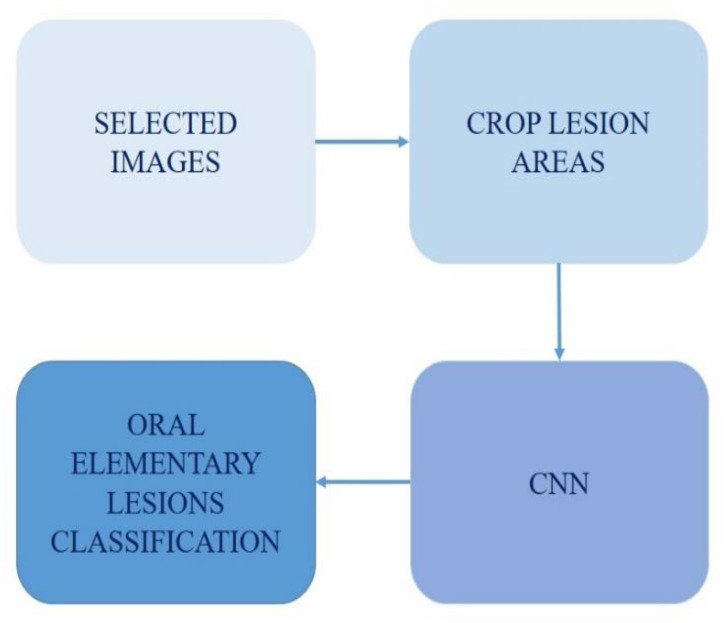
Methodology scheme.

**Figure 2 ijerph-20-03894-f002:**
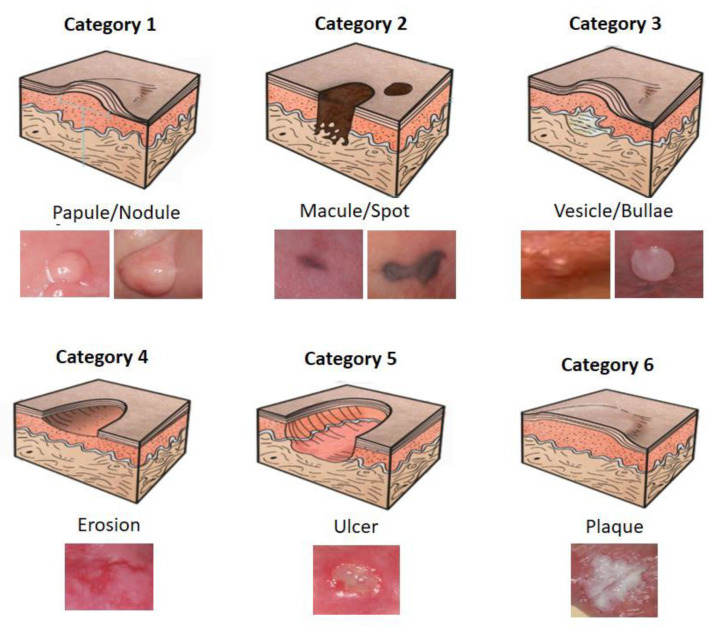
Schematic representation of elementary lesion categories.

**Figure 3 ijerph-20-03894-f003:**
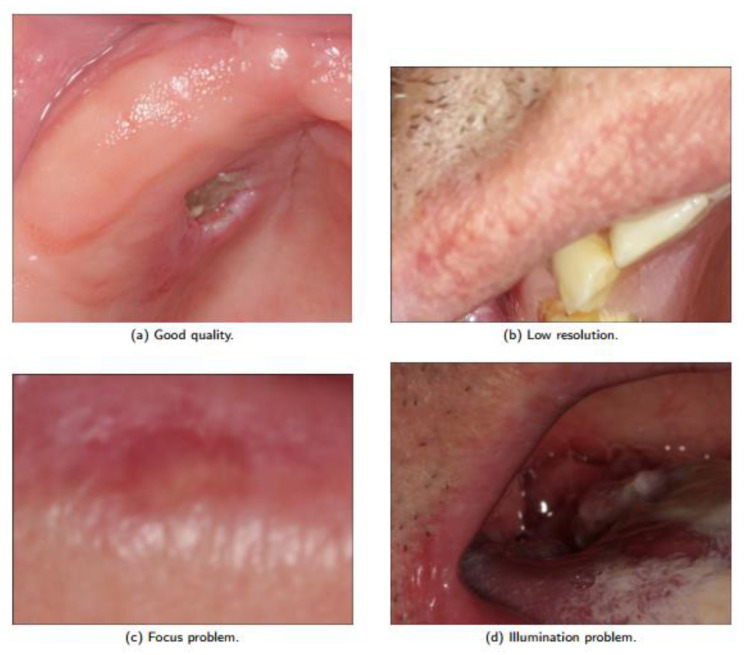
Dataset quality image variation.

**Figure 4 ijerph-20-03894-f004:**
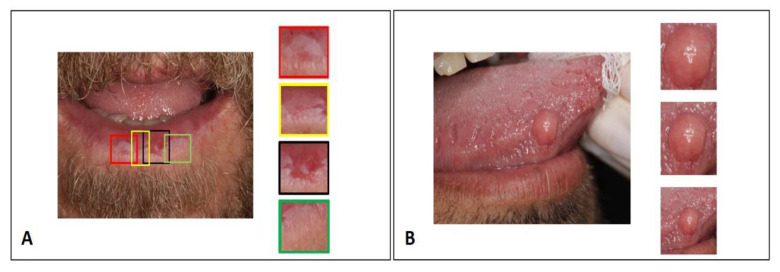
Image crop strategies for network training. (**A**) Images with multiple lesions: each lesion was delimited and cropped separately. (**B**) Demonstrates strategy for including lesion boundaries in the training set.

**Figure 5 ijerph-20-03894-f005:**
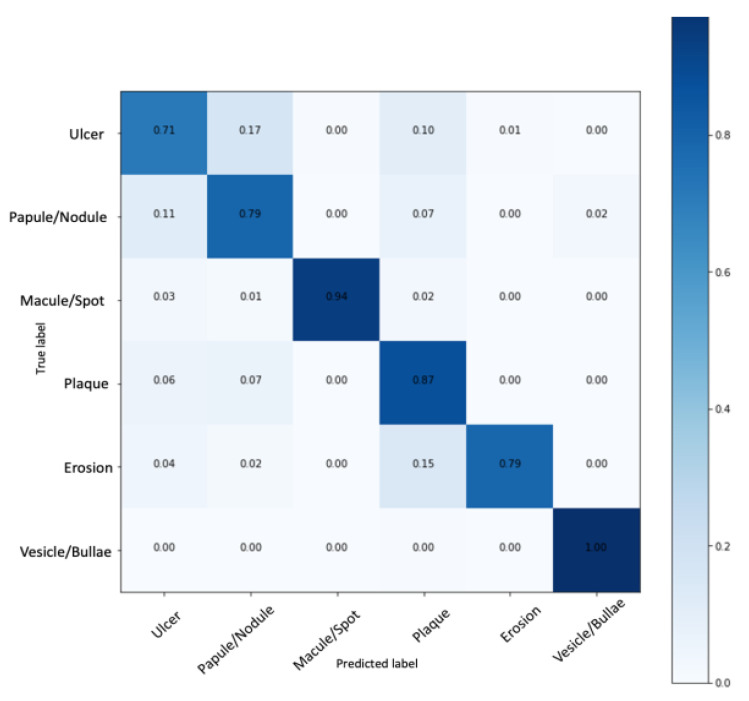
Oral elementary lesion classification using the InceptionV3-based network.

**Table 1 ijerph-20-03894-t001:** Architectures selected for comparison.

Architecture	Size	Parameters	Reference
VGG16	533 MB	138 × 106	[43]
ResNet-50	102 MB	25 × 106	[44]
InceptionV3	96 MB	24 × 106	[29]
Xception	91 MB	23 × 106	[45]

**Table 2 ijerph-20-03894-t002:** Data augmentation of oral elementary lesions.

Class	Number of Images	Augmentation Factor	Resulting Dataset
Plaque	1618	1×	1618
Erosion	159	10×	1590
Vesicle/Bullous	308	5×	1540
Ulcer	1506	1×	1506
Papule/Nodule	1417	1×	1417
Macule/Spot	61	30×	1830

**Table 3 ijerph-20-03894-t003:** Metrics in percentage reported for each of the elementary lesions.

Class	Sensitivity	Specificity	Accuracy	Precision	F1-Score
Ulcer	71.71	95.19	91.30	74.73	73.19
Papule/Nodule	79.79	94.58	92.14	74.52	77.07
Macule/Spot	94.00	100.00	98.99	100.00	96.90
Plaque	87.00	93.17	92.14	71.90	78.73
Erosion	79.00	99.79	96.32	98.75	87.77
Vesicle/Bullous	100.00	99.59	99.66	98.03	99.01

## Data Availability

Data are unavailable due to privacy or ethical restrictions.

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
