# Peer review of "Use of Artificial Intelligence in the Classification of Elementary Oral Lesions from Clinical Images"

_ijerph, 2023, doi:10.3390/ijerph20053894_

Round 1

Reviewer 1 Report

  • What is the resolution of the used images?
  • The authors describe in the methodology that they also used images with different resolutions. How did the different resolutions of the images affect the model's classification results? This is an important element to consider in the results. What were the most low-quality images? Please include information on how many images of what resolution were assigned to which class.
  • Who performed the labeling and outline of the changes? Was it performed by one or several specialists? If several, what was the accuracy of the contouring between specialists? 
  • How many images were ultimately in the network input? 
  • Was cross-validation performed at the stage of splitting into a test set and a training set and subsequently training the network? 
  • How many input layers were in the model? What were the hyperparameters of the convolution layer?
  • In the case of multiclass classification, it will be more important to determine the classification accuracy parameters (ACC, TPR, TNR, etc.) in a macro/micro/weighted version than separately for each class.
  • The innovativeness of the proposed method should resonate better from the beginning when defining the purpose of the work.  
  • In the discussion, it should be added more references to similar solutions and a comparison of the results obtained with the available literature.

Author Response

February 16, 2023

Dear Noah Ye

Assistant Editor

Deputy Assistant Editor. Sq             

Thank you for your time and generous feedback. Your comments and those of the reviewers were highly insightful and enabled us to improve the quality of the manuscript. Below, we provide the point-by-point responses. All modifications in the manuscript have been highlighted in yellow. We hope that the revisions in the manuscript and our accompanying responses will be sufficient to make our manuscript suitable for publication in the International Journal of Environmental Research and Public Health.

We look forward to hearing from you at your earliest convenience.

With kind regards,

The Authors

Reviewer 2 Report

Some doubts and questions:

1) Was the cutting of the rectangles containing the lesions performed by a specialist or a person without knowledge in the health area?

2) The best results were obtained with which neural network architecture and selective hyperparameters?

3) Where are the semantic segmentation results? What are the details of the segmentation methodology?

4) Wouldn't it be possible to compare the results obtained with the results of related works?

5) It would be interesting to include more technical details in the use of neural networks in the methodology, such as the learning rate, number of epochs and all hyperparameters defined (fixed) in the networks used.

Author Response

(The authors gave the same response as above.)

Round 2

Reviewer 1 Report

The article as presented is acceptable for publication.